Baseline seabed habitat and biotope mapping for a proposed marine reserve

Lee Sonny T.M. 1 4 sonny.lee@auckland.ac.nz
Kelly Michelle 2
Langlois Tim J. 3
Costello Mark J. 1
1 Institute of Marine Science, University of Auckland , Auckland , New Zealand
2 National Centre Coasts and Oceans, National Institute of Water & Atmospheric Research , Auckland , New Zealand
3 The UWA Oceans Institute and School of Plant Biology, Faculty of Natural and Agricultural Sciences, The University of Western Australia , Crawley, Western Australia , Australia
4 School of Environment, University of Auckland , Auckland , New Zealand
Dornelas Maria
Electronic publication date: 2015 Dec 10
Publication date: 2015
Volume: 3
Electronic Location ID: e1446
Received 2015 Jul 23; Accepted 2015 Nov 5
Copyright: © 2015 Lee et al.
Copyright year: 2015
Copyright holder: Lee et al.
License: This is an open access article distributed under the terms of the Creative Commons Attribution License, which permits unrestricted use, distribution, reproduction and adaptation in any medium and for any purpose provided that it is properly attributed. For attribution, the original author(s), title, publication source (PeerJ) and either DOI or URL of the article must be cited.
License URL: https://creativecommons.org/licenses/by/4.0/

Keywords: Biodiversity, Benthos, Marine Protected Areas (MPA), Sponges, Fish, Corals

Funding: University of Auckland Research Committee NIWA 2014/2015 SCI This work was funded by a University of Auckland Research Committee grant to Mark J. Costello. MK’s contribution to this research was funded by NIWA under Coasts and Oceans Research Programme 2 Marine Biological Resources: Discovery and definition of the marine biota of New Zealand (2014/2015 SCI). The funders had no role in study design, data collection and analysis, decision to publish, or preparation of the manuscript.

==============================
Seabed mapping can quantify the extent of benthic habitats that comprise marine ecosystems, and assess the impact of fisheries on an ecosystem. In this study, the distribution of seabed habitats in a proposed no-take Marine Reserve along the northeast coast of Great Barrier Island, New Zealand, was mapped using underwater video combined with bathymetry and substratum data. As a result of the boundary extending to the 12 nautical mile Territorial Limit, it would have been the largest coastal Marine Reserve in the country. Recreational and commercial fisheries occur in the region and would be expected to affect species’ abundance. The seabed of the study area and adjacent coastal waters has been trawled up to five times per year. Benthic communities were grouped by multivariate cluster analysis into four biotope classes; namely (1) shallow water macroalgae Ecklonia sp. and Ulva sp. on rocky substrata (Eck.Ulv); and deeper (2) diverse epifauna of sponges and bryozoans on rocky substrata (Por.Bry), (3) brittle star Amphiura sp. and sea anemone Edwardsia sp. on muddy sand (Amph.Edw), and (4) hydroids on mud (Hyd). In biotopes Por.Bry, Amph.Edw and Hyd, there where boulders and rocks were present, and diverse sponge, bryozoan and coral communities. Fifty species were recorded in the deep water survey including significant numbers of the shallow-water hexactinellid glass sponges Symplectella rowi Dendy, 1924 and Rossella ijimai Dendy, 1924, the giant pipe demosponge Isodictya cavicornuta Dendy, 1924, black corals, and locally endemic gorgonians. The habitats identified in the waters to the northeast of Great Barrier Island are likely to be representative of similar depth ranges in northeast New Zealand. This study provides a baseline of the benthic habitats so that should the area become a Marine Reserve, any habitat change might be related to protection from fishing activities and impacts, such as recovery of epifauna following cessation of trawling. The habitat map may also be used to stratify future sampling that would aim to collect and identify epifauna and infauna for identification, and thus better describe the biodiversity of the area.

Introduction

Understanding the spatial distribution of habitats is fundamental to establishing conservation areas and environmental impact assessment, and provides a baseline against which future change in biodiversity can be recognized (Neilson & Costello, 1999; McMath et al., 2000; Costello, 2009; Leleu et al., 2012). Habitat maps provide the spatial structure of ecosystems that is fundamental to understanding biodiversity (Costello, 2001; Costello, 2009; Appeltans et al., 2013). The milieu of habitats not only characterizes an ecosystem, but provides a surrogate for estimating biodiversity where species level surveys are unavailable (Ward et al., 1999; Costello, Cheung & De Hauwere, 2010). Habitat maps have also been used to identify sites that incorporate the ecological processes that support biodiversity, including the presence of exploitable species, vulnerable life stages, and habitat inter-connectivity (Roberts et al., 2003). They also provide the context for biodiversity management which operates at a landscape (and seascape) level (Perrings, Folke & Maler, 1992; Lundblad et al., 2006; Smale et al., 2012; Rovere et al., 2013).

Seabed mapping has improved with technical advances. Habitat mapping studies have been carried out using airborne and satellite remote sensing (Green et al., 1998; Mumby et al., 1998; Malthus & Mumby, 2003; Purkis & Pasterkamp, 2004; Lundblad et al., 2006), single-beam acoustic ground discrimination systems (Foster-Smith & Sotheran, 2003), side-scan sonar and multibeam acoustic (Brown et al., 2002; Kenny et al., 2003), autonomous underwater vehicle (Smale et al., 2012) and underwater digital video (Andréfouët et al., 2008). These techniques vary in cost, resolution capabilities, and the need for data processing and expertise (Buhl-Mortensen et al., 2015). However, aerial and satellite methods only detect very shallow habitats depending on water transparency, so acoustic and drop-down video are the most effective non-destructive methods for waters where scuba diving is not practical (Emblow, Costello & Wyn, 1999; Sides et al., 1995). In the present study, we mapped seabed habitats using video and available data on bathymetry and substrata off the north-east coast of New Zealand.

Almost half of the described marine species around New Zealand are endemic (Gordon et al., 2010), the highest proportion for any country in the world (Costello et al., 2010). The Department of Conservation has a directive to protect a full range of marine habitats and ecosystems that represent New Zealand’s indigenous marine biodiversity (Director-General for Conservation, 2000). There are currently 38 Marine Reserves in New Zealand that fully protect biodiversity (i.e., completely no-take) and act as control sites for studying the effects of fisheries on adjacent areas (Ballantine, 2014; Costello, 2014). In 2004 Marine Reserve status was proposed for the northeast coast of Great Barrier Island (Director-General for Conservation, 2000), but the application was declined by the Minister of Fisheries in 2008. The purpose of this paper is to (1) integrate all previous information on seabed habitats for the entire proposed Great Barrier Island Marine Reserve, including the seashore to deep (>80 m) waters, and (2) to map these areas. The impetus of the present study was to provide a baseline to support research and management should the area become a Marine Reserve (Langlois, Anderson & Babcock, 2006), and provides the first seabed biotope maps below 40 m depth in the region.

Study Area

Great Barrier Island is New Zealand’s fourth largest island and is the largest island off the coast of the North Island (Fig. 1). The surrounding waters range from relatively shallow inshore waters to deeper waters, and include coastal temperate rocky reef and soft-sediment habitats. The East Auckland Current, which is part of the larger South Pacific water circulation, has a strong influence on the coastal ecology of the region and the northeast coast of Great Barrier Island. Previous studies around Great Barrier Island included surveys in the shallower regions, such as on the sub-tidal geology (Moore & Kenny, 1985). Scuba surveys in seven localities in Rangiwhakaea Bay found 63 species of fish, and reported that the dominant habitats were rocky reef with encrusting, turfing, and larger seaweeds including Ecklonia radiata (kelp) and Carpophyllum species (Roberts et al., 1986). Three species of crayfish were more frequent and abundant in the area than on the mainland (Creese & McDowall, 2001). Seaweed communities consisted of three zones, namely shallow mixed weed, encrusting and turfing algae, and kelp; and over 66 species of algae have been reported (Francis & Grace, 1986). Five benthic communities on gravel to muddy-sand sediments from 35 dredge samples were mapped in the northeast corner of the island to approximately 60 m depth (Hayward et al., 1986). Morrison, Drury & Shankar (2001) carried out an acoustic survey of the seafloor habitats to the northeast of Great Barrier Island and provided broad-scale information about the habitats and communities. These studies were limited to shallow waters less than 80 m, describing specific habitats such as soft corals, algal and fish distributions.

Figure 1 The study area to the northeast of Great Barrier Island in New Zealand, southwest Pacific.

The proposed Marine Reserve boundary is shown (solid line). Depth is shown from shallow (red) to deep (blue) with 30, 60, 90 and 120 m depth contours. Dots indicate locations of 119 sampling stations using ROV, BUV, and DDV underwater video.

The area has less recreational fishing than areas closer to the mainland, but such pressure has not been quantified. However, from 2007–2010, the seabed area was trawled up to five times a year (Ministry for Primary Industries, 2012).

Methods

The present study surveyed the area off the northeast coast of Great Barrier Island, between latitude 36.03° and 36.45° south and longitude 175.58° and 176.28° east, approximately 90 km northeast of Auckland, New Zealand (Fig. 1). The area extends from Needles Point in the north to Korotit Bay in the south, and from mean high water spring to 12 nautical miles offshore. Sediment information from the nautical chart (New Zealand Hydrographic Authority, 2012) was digitized to map the substrata. The survey approach was to get an overview of the benthic biotopes for the area of the proposed reserve, and to increase sample density where there was greater physical habitat diversity (e.g., rocks) and decrease effort where previous data existed.

Three benthic surveys were undertaken around Great Barrier Island. The first survey, from 6–9 May 2002 and 17–20 December 2002, was at four rocky sites, using a video camera attached to a remotely operated vehicle (ROV). We updated species identifications from an earlier report of these samples by Sivaguru & Grace (2004). In October 2006, a combination of downward facing and horizontally facing remote underwater video (Langlois et al., 2010) was used to survey along the east coast of Great Barrier Island at 18 nearshore and 12 deepwater sites. Between April 2006 and September 2009, a Drop Down Video (DDV) (Fig. 2) was used to survey 85 stations selected to fill gaps in geographic coverage of previous surveys, and validate depth and sediment information obtained from navigational charts. A total of 119 sites were surveyed in this study. Variations in bathymetry were initially identified using the depth sounder on the research vessel. Then the DDV was used to record the seabed relief, and the presence and relative abundance of the major habitat-forming flora and fauna. The DDV had a color video camera mounted at the base of a pyramidal stand (Fig. 2). The video camera output a PAL composite video signal, approximately 450 horizontal lines, with 120 degree wide angle view. Sampling stations were illuminated by two light housings at both sides of the camera. Each light housing contained three high brightness, Luxeon type LED, with a total output of 1800 Lumen and 125 degree beam angle. The video camera and lights were attached to a waterproof, two-axis rotation device that could rotate the camera and lights 90 degrees vertically and 320 degree horizontally. Images were viewed in real-time aboard the ship so any malfunctions (e.g., leaks, camera fall) could be addressed. Weights were attached to the bottom of the stand to aid stability underwater. The camera and the battery pack were enclosed in a plexiglass container. Once the camera was dropped into the water, the survey began as soon as visibility was sufficient. The camera could rotate 360° to view the habitat. When on the reef edge more than one habitat was recorded. By lifting the camera system off the seabed and drifting the boat, more samples were collected and time was saved by not needing to retrieve and redeploy the system. The video was left on the bottom for approximately 5 min to sample each location. Each sample comprised of the observations of habitat and taxa present at the sampling station. All video cameras used in this study were stationary and sampled a similar area size, to ensure that samples obtained were comparable.

Figure 2 The drop-down video (DDV) system used at most 85 sampling stations in cruise 3 between April 2006 and September 2009.

All video footage was reviewed in the laboratory to identify the percent cover of the substrata and species. The number of specimens at each sampling station was counted. Analysis of species was on presence only as is typical in ecological studies looking across a wide variety of taxa where actual abundances and/or cover are not equivalent, and absences may not be true. The substrata were classified into five different categories—mud, rocks with sediments, sand, and a mixture of rocks and mud based on digitized images (Congalton & Green, 1999). Rocky substrata were aggregations of loose carbonate or volcanic rock fragments. Individual rocks ranged in diameter from 0.25–3 m, gravel 2–250 mm, sand 0.1–2 mm, and mud <0.1 mm. Substrate information was obtained from hydrographic and maritime chart (New Zealand Hydrographic Authority, Chart NZ 5222). Benthic invertebrates (Table 1) were identified directly from DDV video images; no specimens were examined. Species nomenclature follows the World Register of Marine Species (Costello et al., 2013; Boxshall et al., 2015; Van Soest et al., 2015).

Table 1 Species recorded in the present study from images.

Nomenclature follows Boxshall et al. (2015) and Van Soest et al. (2015).

Class	Order	Family	Species	
Bryozoa				
Gymnolaemata	Cheilostomatida	Conescharellinidae	Conescharellina pala Gordon, 1989	
Gymnolaemata	Cheilostomatida	Otionellidae	Otionellina affinis (Cook & Chimonides, 1984)	
Chlorophyta				
Ulvophyceae	Ulvales	Ulvaceae	Ulva sp. indet Linnaeus, 1753	
Cnidaria				
Anthozoa	Actiniaria	Actiniidae	Bunodactis Verrill, 1899 sp. indet.	
Anthozoa	Actiniaria	Edwardsiidae	Edwardsia Quatrefages, 1842 sp. indet.	
Anthozoa	Alcyonacea	Alcyonidae	Alcyonium Linnaeus, 1758 sp. indet.	
Anthozoa	Alcyonacea	Isididae	Keratoisis Wright, 1869 sp. indet.	
Anthozoa	Antipatharia	Antipathidae	Antipatharia Pallas, 1766 sp. indet	
Anthozoa	Corallimorpharia	Corallimorphidae	Corynactis Allman, 1846 sp. indet.	
Anthozoa	Hydroida	Hydrozoa	indet. Owen, 1843	
Anthozoa	Scleractinia	Caryophylliidae	Caryophyllia quadragenaria Lamarck, 1801	
Anthozoa	Scleractinia	Flabellidae	Monomyces rubrum (Quoy & Gaimard, 1833)	
Anthozoa	Scleractinia	Turbinoliidae	Kionotrochus suteri Dennant, 1906	
Hydrozoa	Leptothecata	Plumulariidae	Plumularia Lamarck, 1816 sp. indet.	
Echinodermata				
Asteroidea	Valvatida	Asterodiscididae	Asterodiscides truncatus (Coleman, 1911)	
Asteroidea	Valvatida	Ganeriidae	Knightaster bakeri H.E.S. Clark, 1972	
Asteroidea	Valvatida	Ophidiasteridae	Ophidiaster kermadecensis Benham, 1911	
Crinoidea	Comatulida	Comatulidae	Comanthus AH Clark, 1908 sp. indet.	
Holothuroidea	Aspidochirotida	Stichopodidae	Stichopus mollis (Hutton, 1872)	
Ophiuroidea	Ophiurida	Amphiuridae	Amphiura Forbes, 1843 sp. indet.	
Ochrophyta				
Phaeophyceae	Indet.	Indet.	Indet. Kjellman, 1891	
Porifera				
Calcarea	Clathrinida	Leucaltidae	Leucettusa lancifer Dendy, 1924	
Hexactinellida	Lyssacinosida	Rossellidae	Rossella ijimai Dendy, 1924	
Hexactinellida	Lyssacinosida	Rossellidae	Symplectella rowi Dendy, 1924	
Demospongiae	Astrophorida	Ancorinidae	Ancorina stalagmoidesDendy, 1924?	
Demospongiae	Astrophorida	Ancorinidae	Stelletta crater Dendy, 1924	
Demospongiae	Astrophorida	Ancorinidae	Stelletta maori Dendy, 1924	
Demospongiae	Astrophorida	Geodiidae	Geodia rexDendy, 1924 (rare form)	
Demospongiae	Dictyoceratida	Spongiidae	Spongia gorgonocephalus Cook & Bergquist, 2001	
Demospongiae	Hadromerida	Polymastiidae	Polymastia croceus Kelly-Borges & Bergquist, 1997	
Demospongiae	Hadromerida	Suberitidae	Homaxinella erecta (Brondsted, 1924)?	
Demospongiae	Hadromerida	Trachycladidae	Trachycladus stylifer Dendy, 1924	
Demospongiae	Halichondrida	Halichondriidae	Hymeniacidon sphaerodigitata Bergquist, 1970	
Demospongiae	Halichondrida	Halichondriidae	Axinellidae spp. indet.	
Demospongiae	Haplosclerida	Callyspongiidae	Callyspongia ramosa Duchassaing & Michelotti, 1864	
Demospongiae	Haplosclerida	Chalinidae	Haliclona (Gellius) petrocalyx (Dendy, 1924)	
Demospongiae	Haplosclerida	Petrosiidae	Neopetrosia sp. indet.	
Demospongiae	Haplosclerida	Petrosiidae	Petrosia coralloides Dendy, 1924	
Demospongiae	Haplosclerida	Petrosiidae	Petrosia hebes Lendenfeld, 1888	
Demospongiae	Haplosclerida	Phloeodictyidae	Calyx imperialis (Dendy, 1924)	
Demospongiae	Lithistid Demospongiae	Pleromidae	Pleroma menoui Lévi & Lévi, 1983	
Demospongiae	Lithistid Demospongiae	Scleritodermidae	Aciculites pulchra Dendy, 1924	
Demospongiae	Poecilosclerida	Acarnidae	Iophon laevistylus Dendy, 1924	
Demospongiae	Poecilosclerida	Acarnidae	Iophon minor (Brondsted, 1924)	
Demospongiae	Poecilosclerida	Chondropsidae	Chondropsis kirkii (Bowerbank, 1841) ?	
Demospongiae	Poecilosclerida	Isodictyidae	Isodictya cavicornuta Dendy, 1924	
Demospongiae	Poecilosclerida	Raspailiidae	Raspailia inequalis Dendy, 1924	

The biotope map was digitized using ArcGIS 9.4, and Thiessen’s polygon extension was used to distinguish depth-related patterns in substrata and the different biotope classes. A biotope was defined as a recurring assemblage of species associated with a particular physical habitat (Costello & Emblow, 2005; Costello, 2009). To identify potential species associations and biotopes, samples were (1) compared using Jaccard’s coefficient of similarity on presence-only species data, (2) clustered by species using the group-average method, and (3) statistically significant differences between groups tested using SIMPROF in the PRIMER-E version 6 software (Clarke & Gorley, 2006). Jaccard’s coefficient of similarity was used because it is the simplest similarity coefficient and robust enough to identify species associations.

Results

The 30 m depth contour was less than 1 km, and the 60 m contour about 3 km, from the shoreline (Fig. 1). Rocks and rubble dominated the shallow waters. With increasing depth, sand and then mud dominated the seabed, with occasional rocks. There were frequent rock outcrops with associated ledges and caverns. Beyond 90 m depth, most of the seabed was muddy with occasional patches of rocks and boulders (Fig. 3). In most cases, boundaries between habitats were well defined using the combination of acoustic bathymetry and video surveys. Confident assessment of the current study using the Mapping European Seabed Habitats (MESH) project ‘Remote Techniques Confidence Assessment’ (2004) obtained an overall score of 69 (Table S1).

Figure 3 Cluster analysis.

Dendrogram of results of cluster analysis of samples based on taxa present. The substrata (symbols) and depth (numbers against samples) of each sample are indicated. The four clusters are from the top of the figure, deep-water samples dominated by (A) brittle star Amphiura sp. and sea anemone Edwardsia sp. (Amph.Edw), (B) diverse epifauna of sponges and bryozoans (Por.Bry), (C) hydroids (Hyd), (D) algae (Eck.Ulv).

A total of 66 samples (out of 119 samples) were analysed further for clustering into various biotopes. The other 53 samples that were not analysed are mostly dominated by muddy seabed with no visible signs of epifauna. The 47 taxa identified were clustered into four groups (SIMPROF test, p = 0.05; Table 1, Figs. 3 and 4). When combined with the substrata and depth, there were four distinct biotopes (Fig. 5): (1) shallow water macro-algae, Ecklonia sp. and Ulva sp. on rocky substrata (Eck.Ulv); and deeper water (2) diverse epifauna of sponges and bryozoans on rocky substrata (Por.Bry); (3) brittle star Amphiura sp. and sea anemone Edwardsia sp. on muddy sand (Amph.Edw); and (4) hydroids on mud (Hyd) (Figs. 6 and 7).

Figure 4 Cluster analysis.

Clustering of samples shows four species assemblages: (A) the sea anemone Edwardsia sp. and brittle star Amphiura sp. (Amph.Edw); (B) the diverse epifauna on hard substrata in deeper waters (Por.Bry); (C) hydroids (Hyd); and (D) the kelp Ecklonia sp. and green algae Ulva sp. group (Eck.Ulv).

Figure 5 Cluster analysis.

An alternative presentation of the samples in Fig. 3 using non-metric multi-dimensional scaling (MDS). Vectors show selected (to avoid cluttering plot) taxa indicating the species assemblages. Symbols indicate substrata as in Fig. 3. The four biotopes are indicated by dotted circles: (A) brittle star Amphiura sp. and sea anemone Edwardsia sp. (Amph.Edw), (B) diverse epifauna of sponges and bryozoans (Por.Bry), (C) hydroids (Hyd), (D) algae (Eck.Ulv).

Figure 6 Biotopes matrix.

Map and matrix of the biotopes in the study area off Great Barrier Island, latitude 36.03° and 36.45° south and longitude 175.58° and 176.28° east (land is dark green). Depth contours are in metres. White areas on the map were muddy with no visible epifauna.

Figure 7 Biotopes.

Images of the biotopes found. (A) Shallow (<20 m) rocks covered with encrusting coralline algae, kelp, sponges, corals and bryozoans in biotope Eck.Ulv. (B) Deep (>80 m) mud with sponges and bryozoans growing on any hard substrata in Por.Bry. (C) Brittle star Amphiura sp. and sea anemone Edwardsia sp. on muddy sand in Amph.Edw (D) Deep (>90 m) mud with hydroids and no identifiable epifauna in Hyd.

The rocky-seaweed biotope (Eck.Ulv) occurred in the shallow subtidal, down to a maximum depth of 40 m. Rubble and boulders dominated the sandy seabed. On sand, benthic species such as the sand dollar Fellaster sp. were common. Rubble and boulders were covered in kelp Ecklonia sp. and green algae Ulva sp., and other brown, red and green algae, and epifauna.

Beyond 30 m depth, in biotope Por.Bry, there was a high diversity of epifauna, dominated by sponges and accompanied by gorgonian, corals and bryozoans (Table 1) where there were boulders and hard substrata for attachment. A total of 25 sponge species were tentatively identified from the video clips (Table 1). The number of sponge species identified at each site varied from between six and nine species, and many of the sponge specimens were quite large (Fig. 8). Many more sponge species were seen in the video but were unable to be identified reliably. We identified 25 sponge species with relatively consistent and distinctive morphologies from video clips despite their reduced quality compared to still images. However, sponges with the same general morphology and coloration are not always the same species. For example, there were numerous bowl and fan-shaped haplosclerid species identified in the videos, which may be any one of at least four species: Haliclona (Gellius) petrocalyx Dendy, 1924, Petrosia coralloides Dendy, 1924, Petrosia hebes Lendenfeld, 1888, and Calyx imperialis (Dendy, 1924). The latter three species are very difficult to accurately separate in the field, and even more difficult in images as the morphology of each species is highly variable and interchangeable between species. On the other hand, for species with distinctive morphologies such as Calyx imperialis (Dendy, 1924), which has one or two thin concentrically ridged and veined fans, identification is much easier and more certain if the image is clear. The same can be said for the thick-walled cups and bowls which were identified as species of Stelletta, Ancorina and Geodia spp. Thus, the identifications in Table 1 reflect the current state of our knowledge of species identity in the field from images. At least three black coral species were recorded from the video clips. Dead sections of tree branches that had fallen into the sea were covered in pink jewel anemones Corynactis sp. Brittle stars Astrobrachion constrictum were commonly seen on the living sections of coral. Large gorgonian Keratoisis sp. colonies were also present at some sites.

Figure 8 Glass sponges and demosponges identified from images from deep-reef sites off Great Barrier Island.

(A) Petrosia hebes Lendenfeld, 1888; (B) Hymeniacidon sphaerodigitata Bergquist, 1970; (C) glass sponge Rossella ijimai Dendy, 1924; (D) Aciculites pulchra Dendy, 1924; (E) glass sponge Symplectella rowi Dendy, 1924; (F) Stelletta maori Dendy, 1924; (G) Calyx imperialis (Dendy, 1924); (H) Stelletta crater Dendy, 1924 (left), Spongia (Heterofibria) gorgonocephalus Cook & Bergquist, 2001 (right); (I) Symplectella rowi Dendy, 1924; (J) Iophon laevistylus Dendy, 1924 (left) and Pleroma menoui Lévi & Lévi, 1983 (right); (K) Haliclona (Gellius) petrocalyx (Dendy, 1924); (L) Geodia rex Dendy, 1924; (M, N) Isodictya cavicornuta Dendy, 1924.

The muddy sand (biotope Amph.Edw), dominated by the brittle star Amphiura sp., and sea anemone Edwardsia sp., occurred between 30 m to 100 m depth (Fig. 6). With more hard substrata the previously mentioned diverse epifauna biotope occurred. Thus the distribution of the biotopes depended on depth and substratum (Fig. 6).

Where mud dominated the seabed (biotope Hyd), especially at deeper than 90 m, there were no identifiable epifaunal assemblages, except for occasional hydroids (Fig. 6). However, diverse epifauna biotope could be found when there are occasional rocks in this habitat.

Discussion

The most diverse epifauna were associated with inshore and rocky reef habitats, and some of the area surveyed was deep muddy seabed with no visible epifauna. The substratum and depth formed the primary physical environment of the biotopes (Fig. 6). The  results agreed with and extended previous surveys, indicating that rocky substrata in shallow waters (≤40 depth) were covered by macro-algae, Ecklonia sp. and Ulva sp. (Francis & Grace, 1986; Roberts et al., 1986; Irving & Jeffs, 1992; Kelly & Haggitt, 2002). Most of the deep-water sites surveyed in this study were characterized by muddy sediment. Where boulders rose above the muddy seabed, there was a rich diversity of filter feeders and passive suspension feeders such as hydroids, ophiuroid brittle-star Amphiura sp., and anemone Edwardsia sp.

We recorded 47 invertebrate species, including sponges, gorgonians, black corals and anemones; and deep reef patches that were rich in epifaunal taxa. The two species of scleractinian corals recorded were previously reported on rocky and sediment habitats around Rakitu Island (Fig. 1) below 25 m depth (Brook, 1982). In addition, Brook (1982) reported the scleractinians Culicia rubeola (Quoy & Gaimard, 1833) and Sphenotrochus ralphae Squires, 1964. The former is 6 mm in diameter and encrusts rock crevices in <25 m depth and latter is ca 10 mm and occurred deeper than 25 m attached to shell on sediments. Neither species would be likely to have been observed by our sampling methods.

Sponges

Sponges dominated the invertebrate fauna of the deeper water sites confirming the results of a benthic sled survey in the deep waters off Great Barrier Island (Morrison, Drury & Shankar, 2001).

One of the most important features of the sponge fauna was the presence of several species that are considered rare and restricted in distribution. Abundant specimens of the endemic glass sponge species Symplectella rowi were recorded, this being the first survey to identify a discrete population of the species in Northland waters. A single large specimen of the glass sponge Rossella ijimai was also recorded; this discovery is of particular importance as prior to this survey the species had not been found anywhere in New Zealand waters subsequent to its first description in 1924 (Dendy, 1924; Kelly et al., 2009). These two species were unusual for glass sponges as the majority of species have typically been found in very deep waters on the continental shelf and abyssal plain. Both species have subsequently been found off North Taranaki Bight (Jones et al., 2013).

The presence of several specimens of the giant vase sponge Isodictya cavicornuta Dendy, 1924 is also of considerable significance as this species is rare and has not been collected or recorded since the first description. This distinctive species is known elsewhere only from the Poor Knights where it was last photographed in the 1960s by Roger Grace.

A further feature of the sponge fauna was the presence of numerous large, solid cup-shaped sponges, notably Geodia rex Dendy, 1924, Stelletta crater Dendy, 1924, S. maori Dendy, 1924, Petrosia hebes Lendenfeld, 1888, Calyx imperialisDendy, 1924, Petrosia coralloides Dendy, 1924, and Haliclona (Gellius) petrocalyx (Dendy, 1924).

The sponge fauna shared many species with the North Cape and Three Kings Islands deep-water regions (Dendy, 1924; Kelly et al., 2009), some 100 km further north of the present study area. The large size and abundance of these sponge specimens indicated that some of the community in the deep-reef sites has been relatively undisturbed from trawling and dredging (Tuck et al., 2010; Rooper et al., 2011). However, the Ministry for Primary Industries (2012) data indicated, that the seabed area was trawled up to five times a year between 2007 and 2010, and is still ongoing. It is possible that the absence of visible epifauna in 10% of the area is due to bottom trawling. Other fisheries in the area include angling, spearfishing, scuba diving and potting for crayfish, long-lining and purse seining (Director-General for Conservation, 2000).

Importance of habitat mapping

Benthic ecosystem changes are best quantified with maps of the distribution of biotopes, both to enable detection of natural (including climatic) changes and comparisons between no-take marine reserves and near-by areas that provide understanding of the impacts of human activities. Mapping habitats and biotopes thus aids in the selection of areas as part of a Marine Reserve network (Ward et al., 1999; Thrush et al., 2001; Parnell et al., 2006; Harborne et al., 2008; Dalleau et al., 2010; Bianchi et al., 2012; Rovere et al., 2013), and quantifying fish habitat availability (Ortiz & Tissot, 2008; McLaren et al., 2015). The present study quantified the distribution of biotopes for future comparison on how fishing pressure may modify the Great Barrier Island marine ecosystem. Furthermore, the current study provided a baseline to access the recovery of biodiversity should the sampling area around Great Barrier Island be proposed as a Marine Reserve again.

Although the main purpose of the current study was to investigate the benthic ecosystem in the proposed marine reserve, it would also benefit conservation of fisheries outside the MPA. In a study done by Langlois & Anderson (2006) using similar underwater camera surveys, there was a total of 32 important target and iconic fish species in deepwater and nearshore waters around Great Barrier Island. Some of the most common demersal target species included snapper Pagrus auratus and tarakihi Nemadactylus macropterus. Although there was no evidence that there was any significant difference in the fish assemblage between the proposed marine reserve and comparable nearby areas, the authors predicted that closure to fishing will result in significant increases in fish densities both in the reserves and surrounding waters (Langlois & Anderson, 2006) as has been repeatedly found for reserves around the world (Costello, 2014). Snapper P. auratus predation has major effects on rocky reef habitat structure in coastal marine reserves in northeastern New Zealand, through trophic cascades involving sea urchins and seaweeds (Babcock et al., 1999; Shears & Babcock, 2003), and soft-sediment communities (Langlois, Anderson & Babcock, 2005; Langlois et al., 2006). Therefore, changes in the fisheries around the proposed marine reserve will likely have an impact on the benthic ecosystem and habitat (Langlois, Anderson & Babcock, 2005; Langlois, Harvey & Meeuwig, 2012; Leleu et al., 2012; Langlois & Ballantine, 2005).

In this study, the integration of existing bathymetric and sediment data provided a practical basis for stratifying the sampling using drop-down video. The video adequately identified the physical habitats at all locations and the dominant epifauna. However, in-situ sampling would be necessary to collect invertebrate specimens for species-level identification (Langlois et al., 2006). This is especially the case for infauna in muddy sediments where no epifauna were visible. The present map provides a basis to design a sampling program for collecting invertebrate species to characterize the communities of the study area. This would be desirable if it is again proposed as a Marine Reserve because this information is needed to know what species the reserve would protect, and how representative it would be of the biodiversity of other areas in the region. In addition, this study provided the deepest maps of marine biotopes in New Zealand to date, and discovered new locations of global significance for glass sponge species.

Supplemental Information

Data S1 Raw data for biotope and substrata

Raw data for Great Barrier Island biotope and substrata mapping.

Click here for additional data file.

Table S1 Confident assessment

Confident assessment of the current study using the Mapping European Seabed Habitats (MESH) project.

Click here for additional data file.

We thank Candace Rose-Taylor, Brady Doak, Murray Birch, Agnes Le Port, Jo Evans for assistance in the field, Dr Euan Harvey and Dr Di McLean for lending BRUV equipment, and Dr Kala Sivaguru and Dr Roger Grace for helpful discussion. Murray Birch built the rotating video camera system.

Additional Information and Declarations

Competing Interests

Author Contributions

Data Availability

Mark Costello is an Academic Editor for PeerJ.

Sonny T.M. Lee and Mark J. Costello conceived and designed the experiments, performed the experiments, analyzed the data, contributed reagents/materials/analysis tools, wrote the paper, prepared figures and/or tables, reviewed drafts of the paper.

Michelle Kelly reviewed drafts of the paper, sponges ID and classification.

Tim J. Langlois reviewed drafts of the paper, data and draft review.

The following information was supplied regarding data availability:

The data has been provided as a Data S1.

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
