# Peer review of "Baseline seabed habitat and biotope mapping for a proposed marine reserve"

_PeerJ, doi:10.7717/peerj.1446_

## Round 0.1 · original submission · Major Revisions

Both reviewers praise the quality of this manuscript but highlight areas where it requires clarification. The methods need to be described with sufficient detail so that the validity of the results can be assessed, and, particularly in a paper like this one, so that the methods can be repeated elsewhere.

Reviewer 1 ·

Basic reporting

Overall the English is of a good standard, in places the manuscript lacks clarity and this needs to be addressed (specific comments are attached to the manuscript).
Figure 2 legend: need to add that the 85 sampling sites are from cruise 3.
The figures require labeling to make them easier to interpret:
Figure 3: Label clusters on dendrogram. Need to explain what the red dashed lines are.
Figure 4:Label clusters on dendrogram
Figure 5: Label clusters on nMDS plot
Figure 6: Add some geographic information (latitude/longitude)
Figure 7: No image of biotope 4
Figure 8: Images are no the same size and not all are lined up
Substrate and taxa file require better labeling of rows

Experimental design

The methods are not described with sufficient information that would allow the work to be reproduced. The methods lack detail. For example it is unclear how the samples were collected along a station - where they drift transects (station) with samples taken along them (camera on the seabed for 5 minutes)? If this is the case it needs to be explained much better.
Where the cameras calibrated for area to allow comparison between equipment. This detail is lacking from the methods.
How was substrate grain size measured? i.e were the cameras calibrated?
Need more detail of the statistical methods employed:
- Why was Jaccard’s coefficient used?
- Need to specify that clusters were grouped by species (variables) and not samples

Validity of the findings

In the raw data file there are 66 samples, but it unclear what these represents as there is no reference to this in the manuscript.
It's difficult to tell if the data are robust as it's unclear how the data were analysed - a lot more details are needed:
- How the stations were samples, what constitutes a sample?
- Where non-colonial species enumerated?
The shortcoming of the data should be discussed: the fact that the 3rd cruise includes data from the trawling period - what implication could this have on the data?

Additional comments

The production of the map for the study area is a very useful output, but improvements of the methods and results are needed. While this map is useful you need to be more explicit in the fact that data during trawling periods are used within the analysis.

Annotated reviews are not available for download in order to protect the identity of reviewers who chose to remain anonymous.

Reviewer 2 ·

Basic reporting

This work about seabed mapping in a proposed MPA in NZ is a piece of applied research that is important for its applicability in marine biodiversity conservation and for marine spatial planning purposes.

The paper is very well written.

There are some minor issues: L166: Sentence about substrate needs a citation; L167: I would use "rocks/sediments" instead of "rocks"; Figures: Letters corresponding to each assemblage (A, B, C and D) inserted in the figures 4 and 5 would be of help (Optional).

Experimental design

Although the paper is very well written there are several issues that should be addressed. The first one is the confident assessment. This assessment could be done in several ways but I suggest the authors to follow the one presented by the MESH project (http://www.emodnet-seabedhabitats.eu/default.aspx?page=1635) which is feasible for this case.

Another issue is the need to better clarify how the taxonomic identification was validated, ie, taking that so many taxonomic groups were observed and only based on underwater images how did the authors handled this huge, difficult and complex work?

Furthermore, it seemed that the identification work and the structure of the paper was somehow "sponge" biased. For instance, only sponge species are discriminated in abstract even there are other important species (e.g., black coral and endemic gorgonians?). There is a subchapter in the discussion only dedicated to sponges, rather than the biotopes, and that is not entirely justified (some parts belong to the results section).

Concerning the Drop Down Video (DDV) there are some information that need to be mentioned: image definition, light system, area coverage, positioning and calibration.

Validity of the findings

The results would benefit from a global quantification by species groups clearly showing the species composition of the study area.

Besides, no reference is made to the existence or not of commercial species and that would be of interest in a study like this one. The future MPA could benefit not only the conservation of species and habitats with high conservation value but also the fisheries outside the MPA by spillover effect, for instance.

The use of presence-only species data for the clustering analyses seemed limited and favouring less abundant species and that should be convincingly justified.

Another important aspect is that although is mentioned in the methods section, the analysis to be made in the PRIMER (SIMPROF) is not presented in the results section.

In the discussion, the ecological features that justify the classifying of the study area as a MPA, based on the present study, are not clearly presented.

Additional comments

To conclude, I think that in general this is a valid work that deserves to be published, but only after major improvements.

---

## Round 0.2 · accepted · Accept

All comments raised in the previous review round have been addressed in the revised manuscript.

Reviewer 1 ·

Basic reporting

Overall the manuscript is well written and the much needed clarification has been added.
Minor comments:
Line 59 (and thereafter mentioned), should the code for biotope Pon.Bry be Por.Bry for Porifera?
Line 174, replace ‘at’ with ‘to sample each location’.
Line 221, sub tidal should be one work
Line 235 change ‘colouration’ to ‘coloration’ to be consistent with American spelling used in the manuscript.

Experimental design

All comments have been addressed and now the methods are adequately described to allow repeatability of the work and make use of pre-existing data.

Validity of the findings

The results are valid and are well illustrated with the figures and data provided.

Additional comments

The manuscript is well written, with clarification throughout the manuscript. The applied nature of the work provides maps which can be used for conservation purposes.